# Trust, Transgression and Surrender: Exploring Teacher and SEND Student Perceptions of Engagement with Creative Arts Project-Based Learning (CAPBL) Pedagogies

## James Tarling

Education, School of Society and Culture, Institute of Education, Plymouth University, Plymouth PL4 8AA, UK; james.tarling@plymouth.ac.uk

**Abstract:** CAPBL is an example of a student-led, Creative Arts/Project-Based Learning (CAPBL/PBL) curriculum approach to working with Special Educational Needs and Disabilities (SEND) learners. This mixed-methods, quasi-experimental study seeks to explore staff and student perceptions of working in this way and establish key themes for practice in relation to equity and inclusion. Although the literature on PBL is widespread, CAPBL is novel in that it brings these ideas to a specialist SEND, post-16 context, Further Education (FE), with a particular focus on arts education currently absent from the existing literature. This small-scale research project is positioned as a participant-led action research project involving qualitative/quantitative mixed-method instruments, bassline testing, questionnaires, and semi-structured interviews. Preliminary findings indicate that students and staff experience several benefits to working this way, including positive engagement with learning, self-efficacy, and ownership. This paper attempts to provide workable conclusions for practice located within theoretical frameworks that offer professional resistance to prevailing preoccupations with prescription in curriculum design and pedagogy both nationally and internationally. Specifically, civic compassion and pedagogical partnership are considered in relation to the experiences of learners and staff attempting to work this way. By challenging dominant paradigms of knowledge-led learning at a national level, CAPBL seeks to actively include SEND learners in the global processes of curriculum design itself.

**Keywords:** special needs; education; social inclusion; creative arts; civic compassion; pedagogical partnerships; critical pedagogy





## 1. Introduction

'Letting go is the action of heroes' [1] (p. 164)

When Terry Ellis and Brian Haddow took up their leadership posts at the otherwise unremarkable William Tyndale school in 1974, they may have missed the foreshadowing offered by their benefactors' gruesome history. Tyndale, from whom the school took its name, was a protestant reformer and a biblical scholar who spoke six languages. He resolved to spread the word of the gospel to all people regardless of their station in life and for his progressive views, he was eventually betrayed, publicly strangled, and burnt at the stake. Some four hundred years later, the two men who had instigated progressive regime change at the school towards a form of 'classroom democracy' [2] (p. 275), which included a move away from 'traditional closed classrooms to a more open and flexible system, based on team and cooperative teaching techniques' [3] (p. 11), found themselves similarly at the centre of a community betrayal, rejection, and pillory, with Tyndale labelled the 'School of Shame' [4] (para. 9) in the tabloid press. The affair resulted in a protracted and bitter dispute between parents, staff, leadership, the local authority, and ultimately the government, which culminated in the damning Auld Report [5].

The political and interpersonal machinations of the 'Tyndale Affair' were manifold and complex but in essence involved a pedagogical tug-of-war between two opposing

views of the purpose of education and how schooling should be conducted [2]. On one side, what Claxton [6] (p. XX) would call proponents of a *traditional* direct 'instruction in a knowledge-rich curriculum' (DIKR), and on the other, exploratory *progressives*, in thrall, according to the media and their detractors at the time, to 'a rhetoric of child-centred or holistic education' [6] (p. 1). It has been described as 'a turning point in modern educational history' [2] (p. 275), when:

> Perhaps those teachers, by taking too radical an approach to freedom and choice, opened the door to the new authoritarians and, in doing so set back, perhaps permanently, any prospect of a truly progressive approach to teaching. [4] (para. 14)

This was the moment where it could be argued the wave of progressivism that began with Dewey 'finally broke and rolled back' [7] (p. 39) with 'an endemic culture of performativity' [8] (p. 16) in its wake as, 'the targets, standards and benchmarks become the latest nostrums and settle [. . .] into a complacent and nihilistic orthodoxy' [9] (p. XIV)

The Tyndale affair provides the present day with an allegory of contemporary tensions and transgressions. This article seeks to further explore and discuss a form of 'classroom democracy' [2] (p. 275) in relation to the student and staff experience of progressive pedagogies through their engagement with CAPBL projects devised and co-created by SEND learners in a FE setting. An attempt to promote equity and inclusion through a contemporary iteration of creative, child-centred, and cooperative techniques, it explores staff and student experience of this process and aims to make a novel contribution to a topic largely absent from the specialist SEND FE (post-16) corpus in the UK and internationally.

This article uses Tyndale as a cautionary starting point for a much broader discussion about ways of working in the 'fertile middle ground' [6] (p. 167) between the two polarities of direct instruction and progressive exploration; to make this battleground habitable, particularly for those students at most risk of being disadvantaged by it. There is also an unlikely link to be made between the ebbtide of the Tyndale crisis and UNESCO's Salamanca Conference, widely regarded as a high watermark of debate in relation to inclusive special education [10]; a link which contextualises this endeavour within broader movements in international policy. The unfulfilled promise of educational progressivism, currently in retreat from mainstream education in the UK, can be seen in parallel with the need to protect the aspirations of Salamanca; ideals of education for all of that remain under threat globally from:

> Policy contexts that advocate market forces as a strategy for education, which promote competitiveness, meritocracy and school segregation, under the argument of academic excellence. Extremely disturbing in highly unequal and inequitable education systems. (Duk cited in [10]) (p. 675)

In practical terms, this study seeks to address the issue of unequal access to arts education for disadvantaged learners [11] and explore possibilities for operationalising the benefits of PBL with SEND learners who continue to suffer from academic inequality [12]. PBL as a pedagogy will be explored in more detail in the literature review but can be seen as a learner-led, project-based pedagogy centred around students making, solving, and doing something together with a tangible outcome. This way of working strives to offer alternatives to missed opportunities for involving SEND learners more effectively with their learning by making them active participants in its design through co-creation and partnership [8] so that they may be better placed to face the challenges of transition and exclusion [13].

Although PBL has featured in the literature across international education practice, currently there are significant gaps, both in relation to the arts but particularly in post-16 FE contexts for SEND learners, who are among the most at risk of educational disadvantage. By asking the question, *what is the student and staff perception of engagement, learning, and the experience* of working like this, it is hoped that pedagogies used as part of the CAPBL project can be further developed, reflexively informed by findings from this newly researched and novel setting.

Theoretically, this article will attempt to confront the schism between progressivism and traditional DIKR teaching in an attempt to sow seeds for practice and theoretical enquiry, rooted in a reappraisal of the concept of *risk* as one dimension of a new 'pedagogy of surrender' or teaching as 'letting go', itself a workable articulation of praxis, and alignment with two key theoretical paradigms; that of 'civic compassion' [14] and 'pedagogical partnership' [8] as an attempt to create 'kinder learning spaces' [14] (p. 407) that will nurture socially just and sustainable teaching practice. Practices in which both the teacher and learner are recognised as participants and creators *if* we are willing to acknowledge that, 'education can only be liberatory when everyone claims knowledge as a field in which we all labour' [15] (p. 14).

## 2. Literature Review

This literature review will offer a critical definition of Project-Based Learning (PBL) as the taproot pedagogy from which CAPBL practice grows, whilst reviewing the benefits and qualities associated with this approach. Further to this, pertinent domains of creativity will be explored with reference to SEND education and existing challenges to inclusive and equitable practice. Lastly, a brief examination is offered of the key theoretical literature that informs this study based on the notion of student ownership of curriculum design as a mechanism for encouraging mutually beneficial and inclusive pedagogical partnership [8] and as an articulation of civic compassion [14], which can be applied on a scale outside the UK where this study originates.

PBL is a broad term for a spectrum of practice across all phases and subject specialisms. An early but workable definition is provided by Adderley [16] (p. 1) where PBL involves, 'the solution of a problem', student initiative, a tangible end-product developed over time, and the shifting of teacher input to a more 'advisory rather than authoritarian role', reflecting the pedagogical shift from knowledge transmission to partnership which echo's Sfard's [17] notion of competing metaphors of learning as acquisition or partnership led. In this way, PBL can be seen as located in practices of participation where thought has been given to how teaching and learning may become more democratic; a pedagogical response to what could be seen as 'the harmful effects of treating knowledge as a commodity' [18] (p. 9).

Helpful though this is in aligning the practice of PBL theoretically, for an approach so clearly rooted in classroom practice, a more recent and practically substantial definition is provided by [19] (p. 267), where PBL is, 'characterised by students' autonomy, constructive investigations, goal-setting, collaboration, communication and reflection within real-world practices'. Here, we find evidence of the mechanisms that define PBL and its creative arts bloom. The emphasis on real-world practice is helpful here as art forms made, even at the grassroots level, are made to be experienced in the world at large, 'spread through the whole community' [11] (pp. 7–8). There is a principle at work here which supports the notion that 'students are engaged in working on authentic projects and the development of products' [20] (p. 1) that have counterparts or realisation in the real world.

The premise of PBL is that the traditional teacher/student power relationships are altered to provide the student with more control of the learning journey. From a curriculum design perspective, it can be considered a process rather than a product; the journey is where the learning takes place and 'legitimises the search' [21] (p. 92) for new understanding. The curriculum is no longer controlled centrally but gives, 'sanction and support to open-ended discussions where definitive answers to many questions are not found' [21](p.92). Knowledge, as a single absolute construct, may be contested as learners test assumptions and even teacher expertise on the workbench of real-world experience. Experience that can in turn nurture 'thinking dispositions' and support 'intelligence in the wild' [22] (p. 4).

This has significant implications for young people with SEND in FE; a heterogenous group who are most likely to be excluded or absent from learning and find transition the greatest challenge [13]. This is an under-researched area, especially in the case of specialist post-16 provision, where albeit limited research suggests the conditions may exist in which

SEND learners can be supported to focus on their learning while being treated like an adult [23] (p. 395) by institutions that take seriously, beyond tick-box tokenism, the need to adapt teaching for learners with autism that takes into account 'varying trajectories towards independence' [24] (p. 889). A review of the literature that takes into account these critical concerns of equity, social justice, and inclusion, and reveals a global gap in the writing on the specifics of using PBL/CAPBL approaches with SEND learners in a FE setting, which this article seeks to address.

### 2.1. The Effectiveness of PBL Approaches

Although the literature on the use of PBL and PBL-related approaches in SEND/FE contexts is limited, there is a comprehensive range of study that encompasses the last twenty years across other sectors in UK education. Drawing from that wider corpus of literature, several claims can be considered. Although some Meta-analyses of PBL [19] across multiple settings have highlighted epistemological limitations in some data sets which make it hard to ascertain effectiveness overall, other longitudinal studies [25], which cover a broader range of data and longer timeframes, have noted positive effects on student achievement in comparison with 'traditional' approaches. In addition, the benefits and affordances of PBL in Higher Education (HE) settings in the UK, USA, and Finland, perhaps more closely related in nature to their FE counterparts than mainstream secondary, both in the UK [20] and USA [26], have been identified with their challenges, which will be explored subsequently. FE, often under-represented in research terms, has predictably less evidence to draw on, and Helle and Päivi [27] have criticised the course-specific focus of PBL practices in this sector and the lack of fidelity regarding its nature and implementation.

Examining PBL/CAPBL in terms of its affordances and relationship with traditional, DIKR, instruction-led teaching, there is compelling evidence from Chen and Yang [25] which suggests it could be an effective alternative. However, this highlights a theme present in much of the literature on PBL/CAPBL that whilst acknowledging its value in terms of motivation [28], engagement, and attainment [12], research does not suggest or seek to replace, with any kind of exclusivity, other forms of teaching practice. It is also clear that there are influencing factors that can support or hinder effective implementation. Rather, PBL might be recognised as part of a balanced diet of teaching approaches that may offer a broader, more equitable way of learning for *all* students.

When considering inclusion and equitability from a SEND perspective, the literature that engages directly with the use of PBL with special needs learners is limited, particularly in the UK. However, pertinent studies do indicate several benefits of this way of working including learning gains, motivation, and self-efficacy in terms of positive attitudes towards working with others and improved self-esteem [29]. In addition, when reviewing older international studies [30,31], a picture emerges of a much more widespread adoption of these practices in both special needs and early years contexts, which reveals positive benefits relating to student engagement, learning gains, and inclusion. Some researchers have gone further, describing PBL as 'more effective [...] than the traditional instructional model' [32] (p. 20), that 'projects can promote the effective inclusion of children with disabilities', and 'that teachers of at risk or special needs children had positive feelings about the effect of a project approach when applied in their classrooms [31] (p. 81). These are positive claims but not without critical caveat; many older studies do not refer to UK mainstream or specialist settings and there is a methodological tendency towards small-scale studies. However, the literature reviewed indicates that these limitations do not extend to the diversity of settings and emphasis. There is a pleasing combination of qualitative reportage that engaged with topics such as efficacy and motivation as well as quantitative findings that strive to identify attainment outcomes. Across this vista, a picture emerges of a viable strategy, perhaps somewhat overlooked in the UK and USA in the last ten years outside early years and primary settings, which has the potential for invigorating the SEND classroom for older learners.

*2.2. Professional Challenges in a Context for Creativity*

Mitchell and Rogers [33] identify the professional tension for teachers who may wish to attempt this approach, letting go of the reigns of curriculum and (a) giving students more power over their learning destinations, (b) becoming willing to test their own expertise against unforeseen learning domains presented by the findings their students encounter, and (c) being required to develop greater interpersonal rapport to effectively facilitate this way of working. In a sense, PBL is an approach where the map for the journey is handed over at the beginning and the teacher becomes a guide rather than a leader: 'a collaborator in the learning process rather than a figure of authority or knowledge' [33] (p. 360). Whilst acknowledging these tensions, it has been argued that working with tools such as these has the potential to provide considerable 'possibilities for professional understanding' [34] (p. 60). Proponents of curricula that are driven by the desire to encourage creativity, exploration, and risk taking further describe how this approach engenders an 'observable openness' [34] (p. 58), a willingness to go with the student experience rather than shape it explicitly. In essence, possibility is at the core of project-based learning and although all PBL need not be exclusively associated with creative arts, it is a creative approach in that it requires a degree of creativity from both teacher and student regardless of subject. This can be conceived as a *pedagogy of surrender* that requires teachers to adapt, respond, and take '*flownership*' whilst handing over some control of the curriculum and classroom domains [35] (p. 300).

In relation to the literature on both creativity and project-based learning, there is less to be found on SEND; however, much has been written about the value and potential of creative arts education overall. The literature around this subject highlights a disconnect between the perceived personal value of arts education and its outsider status in national curriculum frameworks [36] and, conversely, the cultural barriers associated with 'high art' which, it has been argued, limits the reach and accessibility of art [11]. These socio-cultural complications suspend the arts in a Newton's Cradle of cultural assumptions and public policy, preventing the very students who might most benefit from gaining regular and sustained access to a more diverse arts-enriched curriculum.

This is concerning given the substantial evidence base that suggests that 'arts participation' can have a positive effect on student progress and attainment whilst potentially supporting the re-engagement of older or disaffected students [12]. This report from the Education Endowment Foundation (EEF) further highlights the value of creative arts participation outside the curriculum and beyond academic attainment. As Carey [11] (p. 255) writes, 'there is evidence that active participation in artwork can engender redemptive self-respect in those who feel excluded from society'. When considering this statement in light of SEN achievement and social inclusion, it is important not to simply equate the SEN student experience with social exclusion and deficit narratives as this overlooks the considerable endeavour, progress, and diversity of the SEN experience overall. However, the EEF's *Special Educational Needs in Mainstream Education Guidance Report* [37] states, 'the attainment gap between pupils with SEND and their peers is twice as big as the gap between pupils eligible for free school meals and their peers' indicating that this is a question of social justice *and* pedagogy.

This identified need for evidence to support tangible practices that encourage equity and inclusion, and the 'leap of faith' [26] that CAPBL asks of teachers can be seen as an articulation of praxis that embodies civic compassion; 'kinder learning spaces that position educators as facilitators of civically engaged and compassionate learning' [14] (p. 407). Through co-creation and letting go of instruction in favour of collaborative adventure, 'giving yourself over to process and getting comfortable with disappearing' [38] (p. 18), compassion is reframed as a 'heart cry' for action:

> 'An action-oriented, critical, and collective response of solidarity to the status quo of neoliberalism, exclusion, and micro and macro forms of inequality as and where they exist'. [8] (p. 22)

Responding to need through the practice of 'pedagogical partnership' [8] (p. 19) that offers a professional and personal, ground-up, rhizomatic 'map of transformations' [39] (p. 602). This 're-wilding' of curriculum ownership encourages students and teachers to work and create together as curriculum authors and 'educational change agents' [40]. Simply put:

> People are happier, more engaged with the world, and more likely to develop or learn, when they are doing and making things for themselves, rather than having things done and made for them. [41] (p. 226)

## 3. Materials and Methods

### 3.1. Positionality and Research Design

This is a small-scale, mixed method, Quasi-Experimental Qualitative (QEQ) pilot study intended to form the first exploratory iteration of a subsequent action research project. This methodology is applicable to settings where random selection and comparison are not feasible or ethical for compassionate/logistical reasons and interventions in real-world, authentic settings are being observed [42,43]. In addition, it was conducted by the author prior to becoming a full-time researcher whilst working as a SENCo and assistant principal, making time constraints and resourcing a complimentary justification for QEQ methodologies [44]. To address the limitations of generalisability and potential bias associated with QEQ, qualitative research instruments were added to the research design to enhance data fidelity by making participant voice a significant component.

The research can be regarded as both participant-led and epistemologically pragmatic in nature; less concerned with defining the nature of reality itself than providing action which can improve practice and address systemic issues in education. Shifting, as it were, 'our understanding of knowledge and the curriculum from the domain of certainty to the domain of the possible' [45] (p. 46). Practitioner research can be said to offer a lens with which to study that is 'problem focused, context specific and future orientated' [46] (p. 5).

However, there are challenges which warrant critical discussion; participants who are also students and colleagues may feel the need to offer appropriate but inauthentic responses. Hence, Denscombe [47] (p. 127) highlights the need to 'be open about the research aspect of their practice. It should not be hidden or disguised'. Conversely, the closeness of the participant to the researcher and the size of the project overall, whilst limiting any claim to objectivity or generalisability, provides narratives that rest in a shared, lived experience. Methodologically, this reflects an honest attempt at transparency regarding the 'messiness' [48] of collaborative practitioner research as a healthy counterpoint to the formal datafication of education and the 'tidy kinds of accountability and measurement associated with quality assurance' [49] (p. 11).

### 3.2. Participant Characteristics and Sample Selection

This study involved a timetabled group of learners who attended a creative arts enrichment session once a week with one qualified drama teacher who devised and ran the session, with a variable rota of three learning support assistants (LSA). Of the five students who participated in the study (presented as students A–E in the data), all students were between eighteen and twenty-two years of age and were deemed to have mental capacity. All participants possessed EHCPs that covered a range of learning needs and complexities: Student B was diagnosed with autism, and Student C with Down's syndrome. Other conditions were present in this cohort including dyslexia; ADHD; and Social, Emotional and Mental Health (SEMH) needs. The gender split was four males and one female.

In order to allow the research project to take place in an authentic educational context where insights into practical effectiveness could be observed, this existing group was selected rather than a randomised group constructed. The group was chosen because the learners had selected this enrichment voluntarily and the member of staff possessed skills and experience related to their subject discipline. Therefore, to reduce disruption

to learning and avoid introducing unnatural or unwelcome transitions, the grassroots suitability and availability of this group was a deciding factor.

### 3.3. Procedures, Ethics, and Methods Used

The study took place over a six-week period to allow for a half term of learning to elapse. It involved five learners, one member of staff, and three Learning Support Assistants (LSA).

At the beginning of the project, consent and ethical approval were discussed with the group and consent forms were completed prior to formal research beginning. One problem encountered was student understanding of ethics, so a brief pre-teaching session was carried out with students to explore and outline key ethical concepts and language: consent, right to withdraw, confidentiality, openness and honesty, and protection from harm. All learners were over eighteen and had the capacity to give consent. All data collected were stored on encrypted devices and held under GDPR guidance at the college.

Surveys were given to learners before and after the project to establish their perspective of learning this way and its impact on their engagement, understanding, and the pedagogies used. At the end of the project, two semi-structured interviews with the lead teacher were conducted to generate supporting quantitative information. Analysis of qualitative data was performed using thematic analysis approaches to establish emerging themes (see Section 5.1) and quantitative data were cleaned, prepared, and analysed for descriptive characteristics [50,51] (see Section 4).

### 3.4. Research Questions and Data Collection

The primary research instrument to address the first research question of *what is the student experience of the CAPBL project in relation to engagement with learning, understanding the topic of self-regulation and the pedagogies used* was a five-point, Likert scale baseline survey and post-action evaluation survey which tracked student responses in relation to their experience of learning in this fashion against the first survey. This included self-assessment questions regarding how well they felt they understood the topic of 'self-regulation' and their own levels of engagement with learning in relation to other classroom-based lessons. The survey also contained an open-ended question about the student experience of working in this way that allowed for broader, qualitative responses.

At the end of the project, two audio-recorded interviews were performed with the classroom teacher to answer the second research question regarding *teacher perception of working with CAPBL*. This was an informal, semi-structured process that began with questions targeted to the research question and the teacher's lived experience of working this way including their perception of student reception and engagement, as well as their own reflections on the key components of this approach that worked/did not work/could be developed.

### 3.5. Creative Arts Project in Practice

The CAPBL project involved a single mixed group of learners from different tutor groups who joined the member of staff for one afternoon per week to do a creative arts vocational session. The session was approximately one and a half hours and the following 'conditions' were agreed upon with the member of staff before embarking on the project:

- The trajectory for the project *should be student-led* based on their ideas and interests.
- There would be a creative arts component which would lead to tangible real-world outcomes *of the students choosing*, e.g., a performance, a video, a document, or an artefact.
- The *curriculum should be seen as a process*; risk taking, possibility thinking, and improvisation were supported within reasonable boundaries of college risk assessment and student individual needs.

The group decided on a project they called 'The Red Zone' based on previous work they had performed with the speech and language team using 'Zones of Regulation' [52], a visual aid that develops emotional literacy by encouraging students to identify, label,

and share how they are feeling. In this context, being 'in the red' denotes feeling angry, overwhelmed, and out of control. They then decided on activities they would perform, effectively writing their own curriculum aims to explore the meaning of The Red Zone both collectively and personally. They agreed their real-world outcome would be a video documentary artefact to capture the process with commentary from the students and footage of a range of activities they devised to explore this concept using creative art. The breadth of the student response was impressive; they scripted and performed to camera for the documentary, and devised transgressive activities to represent heightened emotions including creating ceramic art from plates they smashed, waterbomb games, and other activities.

## 4. Results

Quantitative data were 'cleaned' by checking for errors and inconsistencies [50] and the main characteristics of the data were calculated by comparing the baseline and post-project questionnaires to generate descriptive statistics that outlined [51] student self-assessment.

Qualitative data were analysed and coded using Thematic Analysis (TA), applied with particular reference to the work of Braun and Clarke [53] (p. 79) following 'a method for identifying, analysing and reporting patterns (themes) within data'. In this respect, it is important to consider the limitations of the research; small sample size and a context-specific nature. The aim here is not to draw resolute, scientific conclusions or establish causality but to report and reflect on working patterns of learning as it was lived and experienced by these students and their staff.

In addition to the thematic analysis approach to data analysis, there were questions used in the baseline/post-action evaluation survey which asked for numerical self-assessment scores. This has allowed for some illustrative quantitative analysis. However, it should be noted that due to the small number of students/questions involved, this analysis is only presented to complement and contextually triangulate the experiential themes. A further limitation of this study was the weighting of qualitative data toward the teacher's voice. The interviews yielded some rich commentary, whereas students did not have the same opportunity due to time constraints and access arrangements. Some qualitative responses were generated by aspects of the survey and observations, but these were limited.

Figure 1 provides a summary of the key characteristics of the quantitative student survey data across baseline and post-project.

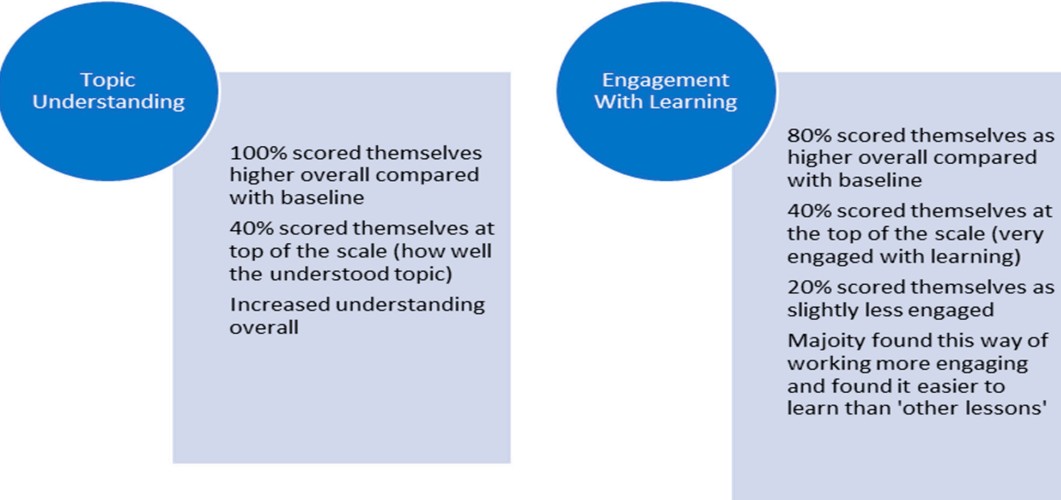

**Figure 1.** Summary of key characteristics of qualitative data from student survey.

Following the headline quantitative data and emerging rounds of TA, a final iteration of analysis was completed which led to three key topics:

1. Trust and Transgression
2. Risk Taking and Teacher Development
3. Leaders and Makers

A digest of key quotes coded and analysed in relation to these topics from both the open-ended student survey question and the teacher interviews are summarised anonymously in Figure 2.

| | Student Responses | Staff Responses |
|---|---|---|
| *(1) Trust and Transgression* | "It's all good". (B)<br><br>"I really enjoyed Smashing plates". (A)<br><br>"Water balloons were great" (A)<br><br>"I wasn't sure at first but got into it." (C)<br><br>'I can't imagine being in the red zone' (E) | "After the practical activities (water balloons, breaking plates) students began to see the fun in the session and relaxed – they then got more relaxed and open to talk about it".<br><br>"Building trust [was important], exploring feeling that were taboo 'spiky and unpleasant' and discuss them more openly, not just with teacher but with each other, expanded sense of collective experience. We were all in it together."<br><br>"Transgression (smashing plates) helped to build trust and reassurance between staff. Sense of fun. Not as just teacher, LSA or student but all of us momentarily 'being children again' - all having fun together." |
| *(2) Risk Taking and Teacher Development* | "Are we allowed to do this?" (C)<br><br>"Is it ok?" (C) | "You're in the moment very much and have to be able to cope with non-linear progression".<br><br>"Keeping the through line in a session takes a lot of mental effort".<br><br>"Embracing change and welcoming the unexpected – needs to teacher to model this."<br><br>"Helped me to have improv/drama-based skills but arguably another teacher might be able to be drawn on different skills". "Having time to explore a concept in a way that makes sense to me – sense of freedom – more exciting."<br><br>"In terms of helping humans talk about feelings that sometimes overwhelm them it was probably as good as it gets for me." |
| *(3) Leaders and Makers* | "I like the making side if it." (A)<br><br>"Makes you happy to feel in charge." (B)<br><br>"Leading it has helped." (A)<br><br>"In normal lessons if it's not going well, I get distracted, but this is fun." (D) | "Really good for developing creativity as part of a balanced approach i.e., nice for it to be a looser, unstructured session one afternoon per week as part of a balanced diet with other more structured sessions."<br><br>"Linking creative 'red work' to daily processes really supported one learner to manage a difficult personal situation and his own 'red zone'." |

**Figure 2.** Qualitative results summary: student open-ended survey response and staff interview response.

## 5. Discussion

### 5.1. Trust and Transgression—Lessons for Student Engagement

An emergent and unexpected theme which arose from the data as a consequence of the co-created choices the students made was the dynamic interaction between notions of trust and transgression as factors which played an important part in the development of the project. This was noted across both student responses and teacher interviews. An illustrative example of this was the triangulation of student feedback which indicated that the majority enjoyed some of the risky/transgressive elements: plate smashing, bubble play, and exploring swearing. All but one student felt they had learned more about their own sense of self-regulation through the process of making, creating, and in some cases destroying.

This indicated a key concept. Supported transgression or risk taking permitted a developmental cycle where *carefully managed* transgression developed greater levels of trust and freedom which in turn gave license to students to open up and engage more directly with the topic at hand. One activity the students devised involved plates being smashed, then reassembled and repainted. This prompted the following feedback which illustrates an expanded sense of classroom experience and renewed boundaries of trust. In relation to the existing literature, there are parallels with the work of Guo et al. [20] and Abrahim and Al-Hoorie [28], which identifies how authentic experiences become more engaging and enjoyable for learners. As the teacher reflected on the risk-taking elements of the project:

> "It wasn't perfect, but it was real!"

This resonates with Craft et al. [54] (p. 551), who identify 'collaborative creativity' taking place in classrooms where there is a shared experience of learning, in this case through possibility thinking and play but also project-based contexts, 'to share their ideas with others and have these recognised' [54] (p. 553). Craft [54] highlights the mercurial nature of risk taking as a component of possibility thinking and creativity, particularly in relation to levels of teacher control and tolerance of either risk or unpredictability. This warrants further research and will be discussed in localised terms in the next thematic section. Importantly, *this study is not advocating risk taking per se* or unbridled excess in relation to managing risk. It has identified a potential link between students feeling safe, developing agency with their learning, and working with trusted staff as a potential component which could promote creative behaviours including risk taking. Furthermore, this collective discovery of learning and the social and emotional coherence it has the potential to engender presents a workable coordinate for pedagogical partnership [8] and civic compassion [14].

### 5.2. Risk Taking and Teacher Development

Quantitative student data present a picture of progress for the majority of students who reported feeling more engaged and improving their understanding of the key concepts (The Red Zone and self-regulation), re-locating earlier findings from international and early years settings in a post-16 specialist SEND context [29–32]. However, to analyse this data critically it needs to be viewed in the context of teacher input and delivery. To what extent does the teacher's contribution, *their* competency, influence the outcome over and above any specific *action* (in this case CAPBL)? A topic for further study would be to what degree this framework for curriculum design could be handed to another teacher with different skills and experience and to what effect?

The teacher in this study had been a professional actor in the past. He was a self-confessed lover of improvisation who had a depth of professional experience to draw on that involved workshopping creative problems, improvising, and operating in scenarios where a pre-planned, desired outcome is *not* the objective.

Their commentary rightly problematises the transferability of this approach but also alludes to the idea that allowing teachers to draw on their own skills to make this approach work could be equally fruitful. As Rogers and Mitchell [33] reflect, teacher confidence

in their competence is key to the success of PBL. This study demonstrates a positive manifestation of this. When it works, the teacher themselves feel engaged and empowered.

This has implications for both further research and teacher development. A next step would be to consider the transferability of CAPBL approaches to other subject areas with staff who possess diverse life experiences and professional skillsets. It also suggests that alternative approaches to staff development could be considered that include working with creative practitioners (actors, musicians, or academics, for example) who can teach improvisation, uncertainty tolerance, and offer toolkits for teaching possibility thinking [55].

### 5.3. Leaders and Makers: Students as Agents of Curriculum Change

In relation to this emerging theme, student voice was strong with the majority of the respondents offering specific comments related to the idea of 'being in charge'. This was an unguarded response to an abstract question that some of the learners in the cohort struggled to answer otherwise. This response does reflect, to some extent, Carey's [11] position that participatory art-based education can engender elements of self-fulfilment, enjoyment, and respect lacking from other parts of the curriculum. It also chimes with Gauntlett's [41] assertions about the value of the pedagogies of making. As the teacher comments:

> "This approach promoted skills and learning behaviours in other areas e.g., willingness to play and explore feelings, be creative, promote dialogue/rapport".

It was an unfortunate and unforeseen misstep of the methods in practice that *did not* allow for more interview time with the learners. The survey data hint that, had more access with the students been available during the timeframe for research, they may have had more to say on this topic.

### 5.4. Next Steps

It is helpful to locate these potentials within a broader framework of policy and practice. Florian [56] (p. 701) argues that post-Salamanca Statement [57] policies at the national and supranational level have persisted in locating special needs at the periphery of mainstream experience when in fact what is needed is a shift in thinking away from the idea of special education as a specialised response to individual difficulty towards one that focuses on extending what is ordinarily available to everyone in the learning community of the classroom.

Furthermore, Ainscow [10] indicates the need for international policy to operate at the classroom level if it is to address the uncertainties and struggles needed to move the principles of inclusion forward. This study suggests that forms of co-created project-based learning where the learner is given greater agency may work towards these aims. As described by the teacher, "not just as teacher, LSA or student, but all of us".

As intimated earlier, there are several preliminary findings which could warrant further research to this end: alternative approaches to staff development, transferability of CAPBL, and exploring the SEN learner voice as leaders of curricula to promote pedagogical partnership and equitable working practices in SEND; practices that are not tethered to a 'deficit orientation of special education discourse' [10] (p. 672).

This action research project lends itself to subsequent iterations in any or all of the suggestions above. For system leaders, it could be argued that there are two pathways to explore: that of curriculum development, which reframes learner choice as paramount, and staff development, which places creativity, improvisation, and uncertainty tolerance at the centre of practice. Additionally, there is perhaps amidst the transparent messiness [48] and limitation of this work an attempt to bring something fresh, even risky, to the field of SEND research in order to address identified gaps in methodology that could be considered collaborative and transformative [58]. Opening the aperture of the research conversation, making space for practitioners to tell 'stories that hold new possibilities for both researchers and teachers and for those who read their stories' [59] (p. 12), acknowledging the entanglements, and diminishing spaces around workload to make that contribution.

In a climate where the craft of teaching can often be reduced to toolkits, interventions, and policy-led implementation [60], it may be a refreshing change for staff development to focus on the art of teaching through participation in similar projects and training with community arts organisations [41]. It is possible to imagine shifting vistas of professional exchange that enhance teaching practice whilst diversifying the parameters of curriculum possibilities to include more partnership organisations and community engagement [35] that reflect the broader intentions of Sustainability Goal Four [61].

## 6. Conclusions

In summary, this small-scale practitioner-led research project, whilst limited in terms of scope and sample size, being a 'study of singularities' and 'accounts of the happenings in single classrooms' [62] (p. 161), has generated initial findings to address gaps in SEND FE research in the UK and internationally to contribute to the growing body of research that recognises the significance of student ownership of learning in the SEND domain [10]. It is the nature of action research that it should be cyclical, and although this is a pilot study encompassing the first phase of this reflexive process, based on these findings, it is recommended further iterations should be implemented and explored in the UK and abroad where PBL is used.

What is clear from the literature and supported by this study is that teachers often feel constrained when it comes to venturing too far off course from their respective curricula, but those that do often return to the path having found something that unexpectedly enriches their curriculum. This unexpected component may be desirable, but it demands a degree of subject confidence and professional self-assurance. To promote this kind of curriculum renewal, a fresh way of approaching staff development is needed that focuses on building skills currently outside the parameters of initial teacher training (ITT): improvisation, uncertainty, tolerance and resilience, possibility thinking, and creativity. Suggestions have been made that this kind of staff development may need specialists from outside the teaching profession to build new partnerships and communities of learning.

Lastly, this study has provided further evidence of how PBL and creative arts may be combined to 'support a more equitable way of engaging' [8] (p. 27) whilst developing students' sense of ownership and belonging as part of a broader theoretical approach related to civic compassion and pedagogical partnership. The literature supporting this in relation to SEND is limited and there is further research recommended in this respect. However, based on these small-scale findings, learners with SEND, much like their mainstream counterparts, found new 'motivational currents' [28] (p. 52) which made learning more engaging. Clearly some, but not all, learners, like the sense of agency this provided and deeper exploration of this idea is recommended.

Overall, this study can be seen as a starting point for discussion around the themes of student ownership and equity, curriculum ownership, inclusion, and holistic conceptions of staff development; re-conceived with theoretical frameworks that seek to redress divisions between direct instruction and exploratory pedagogies by encouraging the 'fertile middle ground' [6] (p. 167) between the two; notions of pedagogical partnership and civic compassion which, it is argued, provide workable coordinates for reimagining how the curriculum is owned and generated. Returning to the Tyndale Affair, results suggest that if staff and schools are willing to tolerate *some* risk to deviate from performative norms and take the 'leap of faith' [26] that this pedagogy of surrender demands, then there is much to gain. Perhaps the greatest potential benefit is the reclamation of sovereignty for SEND students and purposeful professional development for staff, permitting spaces in schools and colleges where vulnerable learners stand to gain the most from pedagogical innovations in an educational climate where too often *education for all*, means 'one size fits no one' [6] (p. XX).

**Funding:** This research received no external funding.

**Institutional Review Board Statement:** The study was conducted in accordance with British Education research Association Guidelines and was approved the Ethics Committee of Plymouth University (March 2022).

**Informed Consent Statement:** Informed consent was obtained from all subjects involved in the study.

**Data Availability Statement:** The data presented in this study are available on request from the corresponding author. The data are not publicly available due to privacy and ethical reasons.

**Conflicts of Interest:** The author declares no conflict of interest.

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
