# Peer review of "Trust, Transgression and Surrender: Exploring Teacher and SEND Student Perceptions of Engagement with Creative Arts Project-Based Learning (CAPBL) Pedagogies"

_education, doi:10.3390/educsci13080848_

Round 1

Reviewer 1 Report

Dear Author, first of all, I appreciate your work. I wish you success in your future studies following this pilot study. I would like to state that your study subject is very valuable and important. However, I find the method section weak. I suggest that the titles: participant characteristics, sample selection, research design, data collection tools, etc. be presented more descriptively and in detail. There is not enough information about the survey and interview form used. It was stated as a qualitative study, but quantitative data are mentioned in the findings, in this case, the method should be explained in more detail. Considering that a 6-week application process and pre-application survey and post-application interviews were made, it can be considered as a quasi-experimental quantitative study. As a result, The method part should be revised and the findings and discussion should be presented in line with the revisions. Kind regards

Reviewer 2 Report

This study presents a novel study that deserves to be published as it is.

Author Response

Dear Reviewer 2,

Thank you for taking the time to review my article and for your supportive and encouraging feedback. I appreciated the positive endorsement very much.

Kind regards

James

Reviewer 3 Report

Thank you for the opportunity to read the manuscript. I really enjoyed reading it and I believe it represents a timely and valuable contribution.

Please conside to make a title short. Title is a lable, not a full grammatical sentence. It should be short and informative.

Please consider  structure as follows:  Introduction, Theoretical framework,  ,Methods, Findings, Discussion 

Your introduction is interesting, but the purpose of introduction is to  inform reader about: Identify what is the overall  problem? Why is it important? What has been done before? What is currently missing ( articulate the gap)?  Objectives – what is the paper doing ( what is my paper offering to fill the gap)? Definitions -what unfamiliar terms  does the paper use ( whuch terms do I need to define -short and informative). It would be to helpful if you state the clear research question, and answer it through your paper. 

Method - the aim of this section i to guarantee validity of  results. Unfortunately this part is very unclear. Please report oe step after the other, describe everything you have done is detail. Who are  participants, and why they  were chosen? Criteria? How you recrute those? Which method you used to achive your objectives? Why? How you developed surveys? Who and how many answered?  How you developed interviews? Who have been interviewed ? How you collected data , and how you analysed the data? Are there any limitation to your methodological approach? Did you have any problems?  How you dealed with the  ethical issues? Overall the methods not correctly exposed and sufficiently informative to allow replication of research.

Results -  The presentations of results is unclear - Where the findings from  intreviews, and where  the findings  from the surveys?    Please describe results precisely. 

Discussion  - The results is presented in the section discusion. The aim of the discussion  is to inform  reader  about the wider meaning and importance of the findings. This is interpritation of the results.  The discussion is not relates to existing knowledge on an international level. 

no comments 
